# Genetics Underlying the Interactions between Neural Crest Cells and Eye Development

**DOI:** 10.3390/jdb8040026

**Published:** 2020-11-10

**Authors:** Jochen Weigele, Brenda L. Bohnsack

**Affiliations:** 1Division of Ophthalmology, Ann & Robert H. Lurie Children’s Hospital of Chicago, 225 E. Chicago Ave, Chicago, IL 60611, USA; jochen.weigele@northwestern.edu; 2Department of Ophthalmology, Northwestern University Feinberg School of Medicine, 645 N. Michigan Ave, Chicago, IL 60611, USA

**Keywords:** neural crest, optic cup, coloboma, microphthalmia, Axenfeld-Rieger Syndrome, Peters Anomaly

## Abstract

The neural crest is a unique, transient stem cell population that is critical for craniofacial and ocular development. Understanding the genetics underlying the steps of neural crest development is essential for gaining insight into the pathogenesis of congenital eye diseases. The neural crest cells play an under-appreciated key role in patterning the neural epithelial-derived optic cup. These interactions between neural crest cells within the periocular mesenchyme and the optic cup, while not well-studied, are critical for optic cup morphogenesis and ocular fissure closure. As a result, microphthalmia and coloboma are common phenotypes in human disease and animal models in which neural crest cell specification and early migration are disrupted. In addition, neural crest cells directly contribute to numerous ocular structures including the cornea, iris, sclera, ciliary body, trabecular meshwork, and aqueous outflow tracts. Defects in later neural crest cell migration and differentiation cause a constellation of well-recognized ocular anterior segment anomalies such as Axenfeld–Rieger Syndrome and Peters Anomaly. This review will focus on the genetics of the neural crest cells within the context of how these complex processes specifically affect overall ocular development and can lead to congenital eye diseases.

## 1. Introduction

The neural crest is a migratory stem cell population which contributes to numerous structures in the anterior segment of the eye, including the cornea, iris, sclera, ciliary body, trabecular meshwork, and aqueous outflow tracts [1,2,3,4,5,6]. The importance of this cell population in anterior segment development is highlighted by examples of potentially blinding diseases such as Axenfeld–Rieger syndrome and Peters anomaly (Figure 1A,B) that are due to genetic defects in neural crest cell migration and differentiation [7,8,9,10,11]. In addition, animal models have demonstrated that neural crest cells have critical cell non-autonomous effects. Disruption of signaling in neural crest cells can lead to alterations in neural epithelial-derived optic cup formation resulting in microphthalmia, anophthalmia, and coloboma (Figure 1C,D) [6,12,13,14]. However, specific interactions between neural crest cells and the neural epithelial-derived optic cup remain undefined.

Neural crest development is a complex process, which is characterized by several landmark events [15,16,17]. Immediately after gastrulation, the ectodermal germ layer is subdivided and specified through different extracellular signaling systems into two territories: the non-neural ectoderm which gives rise to the epidermis and the neural ectoderm, which forms the central nervous system [17,18,19,20]. These territories are separated by the neural plate border, from which neural crest cells are specified during neural tube closure (Figure 2) [17,20,21]. Unique among stem cell populations, neural crest cells then undergo epithelial-to-mesenchymal transition (EMT), delaminate from the neural tube (Figure 3), and migrate to different regions of the embryo to give rise to a broad range of tissues and cells (Figure 4 and Figure 5) [2,22,23]. Each of these successive processes, neural plate border induction, neural crest specification, neural crest migration, and neural crest differentiation is controlled by overlapping gene regulatory networks [15,17,24].

Simultaneous with neural tube closure, the optic vesicle extends from the developing forebrain and then invaginates to form a bi-layered cup (Figure 4) [25,26,27]. Neural crest cells, primarily derived from the prosencephalon and mesencephalon, migrate into the periocular mesenchyme that surrounds and supports the optic cup [10,11,28]. Concurrently, the distal edge of the optic cup interacts with the overlying surface ectoderm to induce lens placode formation followed by separation of the lens vesicle [26,29]. Subsequently, neural crest cells gain entry into the eye via two pathways, 1) the ocular fissure on the ventral edge of the cup and 2) between the optic cup and overlying surface ectoderm to give rise to blood vessels and anterior segment structures [6,11,30].

Although neural crest cells sequentially contribute to the corneal endothelium, corneal stroma, iris, ciliary body, and the aqueous outflow system, the pattern by which these cells migrate into the eye differs between species. The neural crest cells migrate in three waves in humans, in two waves in mice, and in a continuous pattern in chicks [1,4,31]. In zebrafish, neural crest cells migrate in at least three waves. Recent studies have shown that there are two distinct *sox10*-positive cell populations that migrate into the eye 12–18 h after neural crest delamination and a subsequent *foxd3*-positive cell population that migrates into the anterior segment in the late embryonic and early larval stages [11,32]. Following migration into the anterior segment, signals that regulate terminal neural crest cell differentiation are less well-understood and additional work is needed to identify these gene regulatory networks [33,34].

As a stem cell population, the neural crest requires the ability to detect a variety of signals during specific time frames in order to differentiate into a diverse set of tissues and cells. For example, in zebrafish embryos, neural crest cells in the periocular mesenchyme display both overlapping and individualized expression patterns of *foxc1a*, *foxc1b*, *eya2*, *foxd3*, *pitx2*, *sox10*, *lmx1b.1,* and *lmx1b.2* that correspond with the originating neural crest cell migration streams. This reflects a lack of uniformity within the neural crest cell population in the periocular mesenchyme. However, it is a matter of debate whether the neural crest cell population as a whole is homogeneous or a heterogeneous mixture of cells specified toward particular fates [35,36,37,38]. While neural crest development is a complex process involving numerous steps, here we focus on the genetics of this stem cell population in the context of ocular development.

## 2. Overview of Neural Crest Signaling Modules

Neural crest development is divided into successive processes which are regulated by overlapping gene regulatory networks that together form signaling modules [15,17,24,39]. While this review will focus on the genes that have been shown to specifically affect eye development, a brief overview of these signaling modules is required.

Early neural plate border induction is driven by Fgf with the activation of pro-neural genes and is then perpetuated by Notch and Bmp signaling in the neural ectoderm and Wnt in the non-neural ectoderm. Together these signals lead to the induction of neural plate border specifiers (Figure 2) [17,19,20,40,41,42,43]. These transcription factors are initially expressed during early gastrulation in the neural plate border and are critical for neural crest specification. Within the pre-migratory neural crest cells, the expression of the neural plate border specifiers *Msx1*, *Zic2*, and *Tfap2*, is maintained along with activation of additional genes required for proper EMT and subsequent migration of neural crest cells [17,44,45,46]. Those genes can be divided into different groups: 1) CHARGE syndrome-associated genes, 2) genes regulating E-Cadherin in neural crest cell EMT, and 3) genes involved in maintaining embryonic stem cell pluripotency and cell lineage-specificity (Figure 3) [46,47,48,49,50]. The CHARGE syndrome-associated genes regulate cell delamination from the edge of the neural tube via epigenetic modifications. The genes involved in the regulation of E-Cadherin signal the switch from E-Cadherin to N-Cadherin expression, an essential step for preparing cells for migration [47,51,52]. During these processes, however, the cells must maintain their stem cell pluripotency as well as the ability to respond properly to external signals to differentiate into specific cell lineages [46,53]. Once migratory, additional transcription factors direct neural crest cell migration from the edge of the neural tube to destinations throughout the body (Figure 4) [28,54]. Differentiation signals at the end locations, such as the ocular anterior segment, then regulate the formation of the diverse set of structures and tissues (Figure 5) [46].

Common amongst these signaling modules is the repetition of genes in different steps in neural crest development. Furthermore, many of these genes are also expressed in other neural ectodermal-derived tissues. The redundancy of pathways shows the fluidity of these processes but also complicates the determination of critical functions of these genes especially within the context of end-organ development. In the following sections, genes important for both neural crest cell and eye development will be discussed.

## 3. Neural Plate Border Genes

### 3.1. Msx Gene Family

The *Msx* genes encode transcription factors that share homology with the muscle segment homeobox (*msh*) genes in drosophila. As a gene family, they are expressed in diverse tissues throughout embryogenesis, although are best characterized for their role in cranial neural crest cells and thereby craniofacial development [55,56]. The three identified members in the family, *Msx1*, *Msx2,* and *Msx3*, show a functional redundancy where individual gene disruption leads to modest defects while multiple *Msx* gene disruption causes severe abnormalities in neural plate border and neural crest specification [57,58,59,60,61]. *Msx1* and *Msx2* have a continued requirement during specification of the pre-migratory neural crest cells by activating *Arl6ip* and further downstream targets *Snai1/2*, *Foxd3*, *Ets1,* and *Zeb2* (Figure 2) [58]. This pathway leads to the inhibition of E-Cadherin expression, which is essential in triggering EMT [62]. In addition to these early roles, *Msx1* is then expressed in the periocular mesenchyme of mice and chick, while *Msx2* is found later in development in surface-ectoderm-derived corneal epithelium and lens and neuro-epithelial-derived retina [63,64]. However, knockdown of *Msx1* and *Msx2* results in misshapen and enlarged optic cups in mice, but microphthalmia in zebrafish [65]. This phenotypic variation could be due to species differences in the role of *Msx* genes in the forebrain and neural crest cell development. Further studies are required to investigate the detailed function of *Msx1* and *Msx2* during optic cup development and the identification of specific targets of *Msx* genes within the optic vesicle. Abnormal optic cup morphology and colobomas are commonly observed in mice and zebrafish in which expression of genes important in neural crest specification has been genetically disrupted [6,12,13,14]. The mechanisms underlying this interaction and downstream targets within the optic vesicle are further discussed in this review in the context of genes with this shared phenotype.

### 3.2. Zic Gene Family

Members of the *Zic* (zinc finger of the cerebellum) gene family are important regulators of neuroectodermal and neural crest development. While there are five homologs in chick, mice, and humans and seven in zebrafish, *Zic1*, *Zic2* and *Zic3* are the most well-studied [66,67,68]. *Zic1* and *Zic3* work together in neural ectoderm induction which is required for both neural and neural crest cell fates [69,70,71]. However, their primary role, as evidenced by human pathology and animal knockdown and knockout models, is within forebrain development rather than in neural crest-derived tissues [72,73,74,75]. In contrast to *Zic1* and *Zic3*, *Zic2* is a direct target of Wnt-signaling within the anterior neural plate by mid-gastrulation and initiates neural plate border specification [76,77]. In mice, this *Zic2*-regulated step is required for the generation of the appropriate number of neural crest cells [78,79]. Subsequently, in mice and zebrafish, *Zic2* inhibits E-Cadherin via *Nlz1/2* and *Snai1/2* in order to trigger EMT and neural tube exit (Figure 2).

In zebrafish, the expression of *zic2b* persists within neural crest cells while *zic2a* is expressed within a restricted domain in the distal optic stalk [76,77,79,80]. Although the combined *zic2a* and *zic2b* knockout resulted in coloboma with periocular hemorrhage and edema, the main driver of this phenotype was the loss of *zic2b* [79]. Loss of Zic2b results in decreased *alx1* expression and fewer neural crest cells within the periocular mesenchyme, including the *crestin*-negative neural crest cells in the area adjacent to the ocular fissure [79]. In addition, *zic2b* restricts hedgehog (hh) signaling in the neural epithelial-derived pre-optic diencephalon, optic stalk, and ventral retina (Figure 4) [79]. Similarly in zebrafish, *sox4* and *sox11* act by limiting *hh* expression within the ventral forebrain adjacent to the optic vesicle. Furthermore, knockdown of Sox4 or Sox11 results in colobomas in zebrafish [81,82]. While both of these SoxC transcription factors are expressed in the forebrain, *sox4* is also expressed in neural crest cells in the periocular mesenchyme and together with *zic2b* may have a cell non-autonomous effects on optic cup development [82]. Down-regulation of *zic2b*, *sox4*, or *sox11* causes mis-localization of *pax2a*, *vsx1*, and *vsx2*, which are necessary for dorsal and ventral patterning of the retina and subsequent ocular fissure closure [79,81,82]. Although later in development during retinal ganglion cell differentiation, *zic2* antagonizes the expression of *sox4* and *sox11* in patterning axonal projections, the interplay between *zic2* and *sox4* in the neural crest has not been determined [83].

### 3.3. TFAP2 Genes

The *Tfap2* genes encode for the transcription factor AP-2 family members of proteins, which are expressed in both non-neural ectoderm and neural crest cells during embryogenesis [84,85]. The three genes, *Tfap2a*, *Tfap2b*, and *Tfap2c*, show overlapping, but not identical expression within *Xenopus* and mouse neural crest cells, while only *tfap2a* and *tfap2c* are expressed in zebrafish neural crest cells demonstrating species differences in gene function [84,85,86,87,88]. Of the three genes, *Tfap2a* is most critical for neural crest cell and craniofacial development. This is highlighted by the human Branchio-Oculo-Facial Syndrome, which is associated with *TFAP2A* mutations and is characterized by craniofacial (cleft lip, cleft palate, ear malformations) and ocular anomalies (anophthalmia, microphthalmia, cataract, and coloboma) [89,90]. Within the neural plate border, Bmp signaling induces the expression of *Tfap2a* and *Tfap2c* [91]. While *Tfap2a* is a key regulator of *Foxd3* in neural crest specification, *Tfap2c* expression is short-lived and redundant with *Tfap2a* [17,85]. In the pre-migratory neural crest cells, *Tfap2a* expression is independent of *Foxd3*. However, in early migratory cells, *Tfap2a* expression becomes synergistic with and dependent on *Foxd3* and *Sox10* [85,92,93] (Figure 2). Later during migration, *Tfap2a* and *Foxd3* expression segregates into distinct populations accounting for differences in expression of further downstream neural crest markers and mutant phenotypes [92,93]. In post-migratory human, mouse, and zebrafish neural crest cells, *Tfap2a* is expressed in the nasal process, palate, and tooth buds [94]. Similar to *Zic2b*, *Tfap2a* is also expressed in the periocular mesenchyme that surrounds the early optic vesicle in mouse and zebrafish (Figure 4) [94,95]. While *Tfap2a* is expressed in the zebrafish and mouse corneal and lens epithelium and retina, this is after optic cup morphogenesis including ocular fissure closure. Like humans, *Tfap2a* zebrafish and mouse mutants display microphthalmia and coloboma, indicating that neural crest-specific expression has cell non-autonomous effects on the optic cup [94,95]. However, it is unclear whether *Tfap2a* functions through the same hedgehog mediated pathway as *Zic2*.

## 4. CHARGE Syndrome Associated Genes

The CHARGE syndrome-associated genes regulate epigenetic mechanisms to govern the correct delamination of pre-migratory neural crest cells from the edge of the neural tube [96]. *Chd7*, which is central to this signaling cascade, is a helicase DNA-binding protein. In humans, autosomal dominant mutations of *CHD7* are associated with CHARGE syndrome, a constellation of congenital abnormalities including Coloboma, Heart defects, choanal Atresia, Retarded growth and development, Genital hypoplasia, and Ear anomalies [97,98,99]. In mice and *Xenopus*, *Chd7* is expressed in pre-migratory and migratory neural crest cells and is sensitive to gene dosage. Mice heterozygous for *Chd7* mutation or *Xenopus* in which the protein was knocked down with morpholino oligonucleotides showed disorganization and reduction of neural crest cells [48,100]. In neural crest cells, *Chd7* activates *Sox9*, *Twist*, and *Snai1/2* to further direct migration (Figure 3). Additionally, *Chd7* works with the BAF (SWI/SNF) complex, which is responsible for chromatin remodeling. Within the BAF complex, *Brg1* (*Smarca4a*) encodes a core ATPase, and knockout of this gene down-regulates *Tfap2a*, *Foxd3*, and *Snai1/2* expression [100]. In addition, in zebrafish, *chd7* regulates *sema3e*, which is expressed in the hindbrain, and over expression of this *sema3e* rescues the *chd7* knockout phenotype [101]. Consistent with this, mutations in *SEMA3E* have been identified in patients with CHARGE syndrome and knockout of *sema3e* in zebrafish phenocopies *chd7* knockout phenotype.

Colobomas are a prominent feature of CHARGE syndrome, however, it is unclear whether this eye defect is due to neural ectoderm-derived optic cup malformations and/or neural crest defects [102,103]. In addition to its role in the neural crest cells, *Chd7* expression in the neural ectoderm is required for optic cup and stalk development [103]. Hence, additional studies utilizing tissue-specific conditional knockouts are required to investigate the role of *Chd7* in the neural ectoderm and neural crest cells in eye development.

Furthermore, modifications of methylation status by the H3 lysine 4 (H3K4) methylase (encoded by the *Kmt2d* gene) and the X-linked histone H3 lysine 27 (H3K27) demethylase (encoded by the *Kmd6a/Utx* gene) are also important in regulating neural crest cell delamination and subsequent migration [104,105,106]. Mutations in *KMT2D* or *KMD6A* are associated with Kabuki Syndrome, which shares characteristics to CHARGE syndrome including microphthalmia and coloboma [107,108,109]. In *Xenopus*, *Kmt2d* loss of function mutations recapitulate Kabuki syndrome and demonstrate that this methylase is required for dispersion of the migrating neural crest cells [105]. Furthermore, knockdown of Kmt2d in neural crest cells leads to decreased *sema3F* expression in the branchial arches such that over expression of *sema3F* rescues the knockdown. However, this effect is likely indirect as *sema3F* is expressed in adjacent non-neural crest cells and acts as the secreted ligand for the Neuropilin2 (*Npn2)* receptor, which is expressed in the neural crest (Figure 3). [105,110,111]. Similarly, neural crest-specific conditional knockout of *Kmd6a* in mice phenocopies Kabuki syndrome. However, only a small set of targets within those cells are related to histone demethylation [104]. Most prominently within neural crest cells, *Kmd6a* regulates *Chd7* expression and is a target of Notch signaling, the Wnt/β-Catenin pathway, and p53-based targets [112]. While both Notch and Wnt/β-Catenin have defined roles in early neural plate border and subsequent neural crest specification, there may be cross regulation between *p53* and *Kmd6a* as variant *p53* alleles modify the phenotypic severity of *Kmd6a* mutations [112]. Unlike *Chd7* which has roles in both the neural ectoderm and neural crest cells, *Kmt2d* and *Kmd6a* both appear to have neural crest cell-specific functions. Thus, the eye phenotype seen in Kabuki syndrome is likely due to cell non-autonomous effects of the neural crest cells on the optic cup, however, the specific mechanism downstream of these genes that underlie this phenotype has yet to be determined.

### 4.1. E-Cadherin in Neural Crest Cell EMT

Initiation of EMT is a key step in the specification of pre-migratory neural crest cells [23]. The separation of cell-cell interactions and specifically the deconstruction of the tight and adherens junctions between epithelial cells triggers these morphological changes. With dissolution of the adherens junction, E-Cadherin protein is cleaved from the plasma membrane and degraded while E-Cadherin mRNA transcription is down-regulated. This state releases β-Catenin from the plasma membrane allowing it to accumulate in the nucleus and activate transcription in response to Wnt signaling [47,113].

The transcription factor, *Zeb2* (Zinc Finger E-Box Binding Homeobox 2) is an important inhibitor of E-Cadherin in the initiation of EMT in neural crest cells. Expression of *Zeb2* may be regulated by *Msx1* via *Arl6ip1* and *Ets1* [114,115]. *Arl6ip1* encodes ADP ribosylation factor-like 6 interacting protein, which interacts with Arl6 to regulate intracellular trafficking [116]. Interestingly, human mutations in *ARL6* are associated with Bardet–Biedel syndrome, a disease characterized by retinal degeneration, but not neural crest defects [117,118]. However, in zebrafish, knockdown of Arl6ip1 reduces expression of *foxd3*, *snai1b*, *sox10*, and *crestin* and inhibits specification of neural crest sublineages [119]. *Ets1* encodes an E26 transformation-specific transcription factor that targets *Zeb2* in the pre-migratory neural crest cells (Figure 3) [120]. *Ets1* in non-neural crest cells has also been shown to interact with Maf, a basic leucine zipper transcription factor, in which human mutations cause Aymé-Gripp Syndrome [121,122]. This syndrome is characterized by craniofacial abnormalities, sensorineural hearing loss, short stature, developmental delay, and eye anomalies including congenital cataracts and colobomas [122,123]. In mice, *Mafb* is expressed in the neural crest cells and can bind to the *Sox9* promotor [124]. Thus, *Maf* is also likely active within neural crest cells and may act together with *Ets1* to regulate *Zeb2* during neural crest specification.

However, *Zeb2* is expressed in both the neural tube and pre-migratory neural crest cells. In chick, knockdown of Zeb2 causes prolonged maintenance of E-Cadherin in migratory neural crest cells. This disrupts delamination resulting in aggregation of adherent neural crest cells along the neural tube edge [17,125]. Similarly in mice and zebrafish, knockout of *Zeb2* causes neural crest cell migration defects and a phenotype similar to the corresponding *ZEB2* haplosufficient human Mowat–Wilson syndrome [126]. This congenital disease is characterized by neural crest abnormalities (craniofacial and heart anomalies and Hirschsprung disease) as well as neural defects (epilepsy, genesis of the corpus callosum, and developmental delay) [127,128]. While eye abnormalities including coloboma are also present, it is not clear whether these are due to defects in neural and/or neural crest cell development.

*Snai1*/2 encode zinc finger transcription factors that inhibit cadherin expression and trigger EMT. Within the pre-migratory neural crest cells, *Snai1* expression is induced by Wnt signaling and *Zic1*/2, along with *Foxd3*, *Sox10*, and *Ets1* is an early specifier of neural crest cells [45,119,129,130,131]. Wnt signaling regulates *Snai1* expression via the *Tcof1* (Treacle ribosome biogenesis factor 1) gene which is expressed within the pre-migratory and migratory neural crest cells and encodes the Treacle protein (Figure 3) [131]. The Treacle protein resides within the nucleolus and is involved in ribosomal DNA gene transcription. Autosomal dominant mutations in *TCOF1* result in Treacher Collins Syndrome, which is associated with severe craniofacial abnormalities (micrognathia, microtia, midface hypoplasia, zygomatic hypoplasia) and ocular anomalies including coloboma [132,133]. In mice, *Tcof1* is expressed in the frontonasal process and the maxillary and mandibular mesenchyme, and haploinsufficency causes fewer migrating neural crest cells and recapitulation of the human disease [134]. In addition to Wnt signaling, *Snai1/2* is also regulated by *Zic1/2* through the *Nlz1/2* genes, which encode for the NET family of zinc finger transcription repressors [130]. *Nlz1* is expressed on the neural ectoderm side of the neural plate border and appears to demarcate the neural crest cell delamination line [135,136]. Through this action, *Nlz1* indirectly signals via *Snai1/2* the pre-migratory neural crest cells which should undergo EMT [135].

*Snai1* represses E-cadherin within the pre-migratory cells and continues to be expressed in the migratory cells, including in the periocular mesenchyme [137]. *Snai2* is induced by Wnt-signaling following neural plate border specification and represses Cadherin 6b in the pre-migratory neural crest cells along the dorsal neural tube (Figure 3) [138]. Cadherin 6b, like E-Cadherin, needs to be down regulated in order to initiate temporal and spatial emigration of the neural crest cells from the edge of the neural tube [139]. *Snai1* mutants display colobomas, and the specific effects of this gene in regulating EMT suggests that this action in the neural crest cells indirectly affects ocular fissure closure [140]. In contrast, *Snai2* mutants do not appear to have eye abnormalities [141]. This may be related to either the lack of *Snai2* expression within the periocular mesenchyme or compensation of function by *Snai1*. Additional studies are required to further understand the role of the *Snai* genes specifically on ocular development.

### 4.2. Maintaining Embryonic Stem Cell Pluripotency and Cell Lineage-Specificity

Despite acquiring mesenchymal properties and ultimately migratory capacity, the neural crest cells need to remain in a pluripotent state, as well as to be governed towards a specific cell fate. While neural plate border specifiers show continued expression, additional factors, including *Sox10*, *Foxd3*, *Snai1/2,* and *Ets1* are activated at this stage to maintain the stem cell qualities of the pre-migratory neural crest cells [53,142]. However, during migration and post-migration, these transcription factors guide the segregation of sublineages, and eventually the loss of expression triggers terminal differentiation.

*Sox10* encodes for a SRY-related HMG box transcription factor that shuttles proteins between the cytoplasm and nucleus in order to regulate cell fate [143,144]. During migration, *Sox10* is one of the earliest specific markers for undifferentiated neural crest cells, however, in post-migratory cells, *Sox10* directs cells toward melanocyte and glial differentiation [143,144,145,146]. In the zebrafish and mouse craniofacial region, *Sox10* is expressed within the pharyngeal arches and in the periocular mesenchyme [145,147,148,149,150]. In zebrafish, *sox10* continues to be expressed within the mandibular and maxillary cartilage during the early larval stage but is absent in juveniles [11]. While *sox10* positive cells enter into the developing eye, its expression is short-lived and it is unclear to which ocular tissues these neural crest cells contribute [11,32]. In humans, autosomal dominant nonsense and frame shift mutations in *SOX10*, are associated with Waardenburg syndrome, which is primarily characterized by pigmentation abnormalities and mild facial dysmorphism [151,152]. Similarly, the colorless zebrafish mutants, which were generated through an ENU mutagenesis screen, have various point mutations in the *sox10* allele resulting in pigmentation abnormalities, However, the mutant fish show normal craniofacial and ocular development and survive to adulthood [147]. In contrast, morpholino oligonucleotide knockdown of Sox10 in zebrafish disrupts early neural crest specification resulting in significantly fewer migrating cells. Severely affected animals have severe craniofacial abnormalities, the eyes are microphthalmic with colobomas and do not survive past hatching (unpublished data, personal observation). Furthermore, homozygous *Sox10* mutations in mice are lethal in embryonic or early perinatal stages [153,154]. These phenotypic variations may be due to the effects of gene dosage and the amount of residual activity in the heterozygous versus homozygous states. Additional studies are required to further investigate these discrepancies and the role of *Sox10* in neural crest cells and optic vesicle development.

*Foxd3* encodes a forkhead winged-helix transcription factor, which is involved in changing stem cells from a naive state to a primed pluripotent state. Foxd3 works by modifying chromatin structure to either decrease recruitment or repress activation of enhancers [38]. Like *Sox10*, *Foxd3* is an early marker for neural crest cells but directs post-migratory differentiation towards neuronal, bone, and cartilage fates [155,156]. *Foxd3* is also expressed within the pharyngeal arches and periocular mesenchyme [30,92]. In zebrafish, *foxd3*-positive cells enter into the developing eye later than the smaller population of *sox10*-positive cells. During the embryonic into early larval stages, *foxd3*-positive cells populate the iris and cornea, however, expression is down-regulated by mid-larval stage [11]. A variant allele of human *FOXD3* has been associated with pigmentation abnormalities (vitiligo). However, this mutation, which is located within the promoter region, has been shown to increase transcriptional activity in vitro and may also be involved in autoimmune processes such as Hashimoto’s thyroiditis [157]. To date, no haploinsufficient mutations of *FOXD3* have been identified in human diseases. This correlates with the animal *Foxd3* knockout and knockdown models, which show severe neural crest defects especially involving the peripheral nervous system and cartilages that are incompatible with survival. In zebrafish, morpholino oligonucleotide Foxd3 knockdown causes microphthalmia with impaired neural crest cell migration into the eye [158,159,160,161]. Interestingly, one clinical study showed that *FOXD3* variants of unknown pathogenic significance were identified in a handful of patients with aniridia and Peters Anomaly [162]. Nonetheless, further investigation is required to determine the role of *Foxd3* in eye development and specifically in the regulation of neural crest cell interactions with the optic cup.

The *Alx* family of genes encodes for paired-class homeobox transcription factors, and all three members, *Alx1*, *Alx3*, and *Alx4* play various roles in neural crest development [163]. *Alx1* coregulates *Sox10* and *Foxd3* during early neural crest cell specification and continues to be expressed throughout migration. In addition, *Alx1* is expressed in the periocular mesenchyme [164]. In humans, mutations of *ALX1* result in frontonasal dysplasia and eye abnormalities including microphthalmia and anophthalmia [165,166]. Correspondingly, knockout of *alx1* in zebrafish also causes microphthalmia and coloboma. In contrast, *alx3* and *alx4* play later roles in post-migratory neural crest cells and are only expressed in limited areas of the periocular mesenchyme and eye [164]. Subsequently, human mutations in either *ALX3* or *ALX4* cause mild frononasal dysplasia, but no ocular abnormalities [167,168]. Thus, early disruption of neural crest cell specification and maintenance of pluripotency by *Alx1* is required for normal optic cup development and ocular fissure closure. This effect of *Alx1*, is likely due to cell non-autonomous effects on the optic cup mediated through neural crest cells. However, downstream targets and molecular events need to be further defined.

## 5. Cranial Neural Crest Cell Migration Gene Regulation

After delamination from the edge of the neural tube, neural crest cells follow distinct pathways throughout the body. The cranial neural crest cells originate between the diencephalon and the rhombencephalon to eventually contribute to much of the craniofacial skeleton (frontal, nasal, mandible, maxillary, zygomatic), and numerous eye structures (corneal stroma, corneal endothelium, trabecular meshwork, sclera, iris stroma, uveal melanocytes, ciliary body muscle) [6,34,169,170,171]. Cranial neural crest cells which originate from the diencephalon and anterior mesencephalon migrate in streams dorsal and ventral to the eye to populate the periocular mesenchyme and frontonasal process [11]. Cells from the posterior mesencephalon and rhombencephalon successively migrate into the pharyngeal arches [11,172]. Within these streams of cranial neural crest cells, molecular markers are already distinguishing into subpopulations. *Crestin*, a well-defined marker for migratory neural crest cells is expressed in the dorsal, but not ventral periocular mesenchyme [173]. In zebrafish, while there is initially coexpression of *sox10* and *foxd3* during early migration into the pharyngeal arches and periocular mesenchyme, the timing of the down regulation of these transcription factors varies depending on the triggering of terminal differentiation [11,174]. During this process, the expression of additional transcription factors including *pitx2*, *foxc1*, *eya2*, and *lmx1b* is upregulated in cranial neural crest cells in order to regulate migration to final destinations and differentiation (Figure 4) [174].

*Pitx2* encodes for a paired-like homeodomain transcription factor first identified within the pituitary as a regulator of prolactin. However, extensive animal studies have shown *Pitx2* to be critical for establishing left-right axis and subsequent asymmetrical development of the heart, lungs, and gastrointestinal system [175,176,177]. Thus, complete knockout of *Pitx2* in mice is embryonic lethal due to severe cardiac abnormalities. Nevertheless, these mutants displayed severe eye abnormalities including microphthalmia and coloboma [175,176]. Tissue specific-knockouts demonstrated that these ocular defects are due to cell non-autonomous effects of *Pitx2* within neural crest cells on optic cup development [178]. In humans, autosomal dominant mutations in *PITX2* are most commonly associated with Axenfeld–Rieger Syndrome, which consists of a distinct set of systemic, craniofacial, and ocular anomalies that stem from disruption of neural crest cell migration and differentiation. Systemic abnormalities include Hirschsprungs disease (absence of enteric innervation of segments of intestine) and cardiac outflow tract anomalies while mid-face hypoplasia with oligo/microdontia are the most common craniofacial findings. In addition, extraocular muscle and eyelid abnormalities have been associated with Axenfeld–Rieger Syndrome [179,180]. Within the eye, there are bridging strands that connect the iris to an anteriorized Schwalbe’s line, which demarcates the outer edge of the corneal endothelium. In addition, the iris is hypoplastic resulting in corectopia (irregular pupil) and pseudopolycoria (multiple pupils) [181,182,183,184]. Interestingly, in a smaller subset of patients, optic nerve abnormalities including coloboma and morning glory appearance are present (unpublished data, personal observation). The greatest source of morbidity is vision loss due to glaucoma that affects approximately 50–75% of patients and is often refractory to medical treatment [185]. Less commonly, *PITX2* mutations are associated with Peters Anomaly, which is characterized by a central corneal opacity that is due to incomplete separation of the lens vesicle from the corneal epithelium and failure of neural crest cells to migrate into the anterior segment [186].

*Pitx2* is first expressed in neural crest cells after migration has been initiated and together with the morphogen, retinoic acid (RA), directs the migrating neural crest cells along different streams into the pharyngeal arches and periocular mesenchyme [176,187]. Knockdown of Pitx2 in zebrafish did not alter the number of delaminating neural crest cells but disrupted migratory pathways into the craniofacial region resulting in apoptosis of these cells [10]. This resulted in mandibular and maxillary malformations as well as ocular anterior segment dysgenesis, coloboma, and microphthalmia [10,176]. *Pitx2* knockout mice and mice in which RA fails to promote *Pitx2* expression (i.e., knockout of multiple RA receptors), show a more severe eye phenotype. This includes anophthalmia, microphthalmia, and defects of the optic nerve such that the eyes attach directly to the ventral hypothalamus [175,178,188]. Conditional knockout of *Pitx2* within the neural crest shows a similar phenotype, indicating that these ocular defects are due to cell non-autonomous function of *Pitx2* in neural crest cells [178]. *Pitx2* expression in the periocular mesenchyme activates signals that cause the morphogenetic extension of the optic stalk such that failure of this step results in a foreshortened optic stalk and microphthalmia/anophthalmia. Subsequently, *Pitx2* expression is predominantly maintained within the periocular mesenchyme where it is required for corneal differentiation [178]. In mice, knockout of *Pitx2* may cause incompetence of the ocular neural crest cells to form corneal endothelium and stroma resulting in malformed and thickened corneas. [176,178,189]. Importantly, *Pitx2* integrates input from RA and Wnt signaling to trigger ocular neural crest cell migration and differentiation (Figure 4) [176,187]. RA, which is produced by the dorsal and ventral retina, targets the periocular mesenchyme via RA receptors (RAR) α and γ [34,176,190,191,192]. Within these periocular neural crest cells, RA directly upregulates the expression of *Pitx2* through binding to the RA response elements. Furthermore, while the Wnt pathway does not initiate activation, the signaling components, β-catenin, and *Lef1*, bind to the *Pitx2* promotor and are subsequently required for maintaining *Pitx2* expression. However, *Pitx2* targets *Dkk2*, an extracellular antagonist that moderates Wnt signaling activity, to form a regulatory feedback loop within the periocular neural crest cells [187,193,194].

Additional downstream targets of *Pitx2* includes *Lmx1b*, a LIM-homeodomain transcription factor that is expressed in periocular mesenchyme and is later enriched in presumptive anterior segment structures (iris, cornea, trabecular meshwork), the ocular fissure, and the hyaloid vasculature (Figure 5) [195]. *Lmx1b* regulates migration and survival of neural crest cells in the anterior segment and *Lmx1b* mutant mice have microphthalmic eyes with iris and ciliary body hypoplasia [195]. In humans, autosomal dominant mutations in *LMX1B* cause Nail–Patella Syndrome which is strongly associated with open-angle glaucoma [196]. However, unlike Axenfeld–Rieger syndrome, affected individuals do not exhibit congenital ocular abnormalities, which suggests an ongoing requirement for the *LMX1B* gene in post-natal maintenance of the structure and function of the trabecular meshwork and sclera. Interestingly, as with Pitx2, morpholino oligonucleotide knockdown of Lmx1b in zebrafish causes colobomas indicating a cell non-autonomous role for these two factors on optic cup development [197]. Another *Pitx2* target within neural crest cells is *Tfap2b* (Figure 5). In contrast to *Tfap2a*, which is important in neural border induction, *Tfap2b* is expressed in the periocular mesenchyme in mice and plays a *Lmx1b*-independent role in anterior chamber development [198]. Neural crest-specific *Tfap2b* knockout mice show mal-developed iris, cornea, trabecular meshwork, and ciliary body resulting in a closed angle-configuration. Loss of *Tfap2b* expression causes absence of N-Cadherin in the corneal endothelium and disorganization of junctional complexes and corneal edema. Furthermore, these *Tfap2b* mutants also have decreased expression of the cornea-specific collagen, *Col8a2*, and the corneal endothelial marker, *Zp4* leading to decreased cell proliferation [199]. While *Lmx1b* and *Tfap2b* have both been identified as *Pitx2* targets, additional studies are required to identify additional downstream factors within the anterior segment.

Although strabismus and blepharophimosis are rarely reported in Axenfeld–Rieger syndrome, in animal models periocular neural crest cells are essential for extraocular muscle formation and organization [179,180,200]. Both RA and *Pitx2* regulate expression of muscle-specific transcription factors such as *Myf5, Myog,* and *Myod1* that are required for extraocular muscle myogenesis and survival [201,202]. Furthermore, *Foxl2* in periocular neural crest cells activates expression of smooth muscle alpha actin (encoded by the *Acta2* gene) in extraocular muscles in mice, and human mutations in *FOXL2* are associated with blepharophimosis syndrome. While *Myf5*, *Myog*, and *Myod1* induce myoblast differentiation and eventual *Acta2* expression, a direct connection at the molecular level between *Pitx2* and *Foxl2* in the periocular mesenchyme has not been established. In contrast, Pitx2 knockdown in zebrafish, while affecting the jaw musculature, did not alter expression of *myod* nor organization of the extraocular muscles. However, the ocular phenotype in *pitx2* knockout mice causes anophthalmia or severe microphthalmia, which is worse than in zebrafish or humans. The eye itself is known to be important in regulating extraocular muscle development such that genetic, toxic, or surgical loss of the optic vesicle causes disorganization and in some cases agenesis of the extraocular muscles [200,203]. Thus, the extraocular muscle phenotype seen in *pitx2* knockout mice may be directly or indirectly related to neural crest cell signaling.

*FoxC1* encodes for a forkhead transcription factor, that functions in craniofacial and ocular neural crest cell migration and differentiation [204]. Like *Pitx2*, *Foxc1* is not required for delamination, but critical for survival and migration of neural crest cells [205]. *Foxc1* is then expressed within the neural crest cells of the periocular mesenchyme and later within the presumptive anterior segment structures [204]. Furthermore, autosomal dominant mutations in *FOXC1* also primarily cause Axenfeld–Rieger syndrome [9,206]. In *Foxc1* knockout mice, the cornea is thickened and disorganized, and there is failure of the lens to separate from the cornea, which is more reminiscent of Peters Anomaly [207]. Knockdown of the zebrafish homolog Foxc1a also results in maldevelopment of the cornea and iris and causes optic nerve hypoplasia and colobomas. In addition, the zebrafish show cerebral hemorrhages suggesting a separate role of Foxc1a in blood vessel integrity. Correspondingly, patients with *FOXC1* mutations have also been found to have early-onset cerebrovascular disease [205].

*Foxc1*, like *Pitx2*, is regulated by RA within the periocular mesenchyme. Furthermore, *Foxc1* and *Pitx2* are coexpressed within neural crest cells and have been found to regulate each other’s expression. In addition, the transcriptional regulator protein, *Pawr* (PRKC apoptosis Wilm’s tumor 1 regulator) modulates the abilities of *Pitx2* and *Foxc1* to activate target genes [193]. *Tgfβ2* signaling from the lens which targets its receptor (Tgfbr2) in the periocular mesenchyme also regulates *Foxc1* and *Pitx2* expression and interestingly, *Tgfβ2* knockout mice share a Peters Anomaly-like phenotype similar to *Foxc1* knockout mice [208,209,210]. However, the main effect of *Tgfβ2* signaling is likely post-migration in directing *Foxc1* and *Pitx2* positive cells toward corneal endothelial and stromal differentiation.

Within neural crest cells, *Foxc1* has been shown to target genes including *Eya2*, *Ffg19*, *Foxo1a*, *Foxc2*, and *Galnt4* (Figure 5). *Eya2* is a transcription factor that is expressed in the periocular mesenchyme and ocular fissure in mice and zebrafish. *Eya2* expression, which is increased by *foxc1a*, RA, and *nlx1*, promotes remodeling of the periocular mesenchyme by inducing apoptosis of neural crests cells [211,212,213]. In presumptive ocular tissues, Foxc1 binds to the *Fgf19* promoter, activating gene transcription which directs both the development and maintenance of anterior segment structures via ERK1/2 signaling [214]. Foxc1 also has been shown to regulate the forkhead transcription factor *Foxo1a*. Within zebrafish and cultured human trabecular meshwork cells, decreased *Foxc1* reduces *Foxo1a* expression, which impaired response to oxidative stress and ultimately survivability [215]. In mouse periocular mesenchyme, *Foxc1* together with *Pitx2* and Wnt-signaling interact with *Foxc2* to cooperatively regulate early corneal development. *Foxc2* is required within neural crest cells to promote corneal cell identity and demarcate corneal and scleral tissues [216]. However, it has yet to be shown whether *Foxc2* functions in the same role in human eye development. *Galnt4* encodes an enzyme which initiates mucin-type O-linked glycosylation and ultimately changes the gel-like properties of mucin secreted from epithelial cells [212]. Although the specific function of *Galnt4* within the neural crest cells is unknown, the involvement of glycosylases in altering extracellular matrices and mediating adhesion, suggests a role in promoting migration. Taken together, *Foxc1* has a variety of targets within the neural crest cells and eye. However, additional studies are required to identify additional effectors and determine how these genes work together to regulate neural crest cell migration and differentiation.

## 6. Summary/Conclusions

Neural crest development is a complex process that involves multiple steps and numerous gene regulatory networks [15,17,24,39]. Genetic disruption can trigger a cascade of downstream effects that can lead to widespread systemic, craniofacial, and ocular abnormalities. Eye development in particular is exquisitely sensitive to proper neural crest cell development.

While microphthalmia, anophthalmia, and coloboma are typically associated with disruption of neural epithelial gene function in the forebrain and optic cup development, human disease and animal models have demonstrated that perturbation of neural crest specification or migration often results in a similar phenotype [6,12,13,14]. As discussed in this review, this common phenotype of microphthalmia and coloboma is exhibited when there is genetic disruption of neural plate border genes (*Msx1*/*2*, *Zic2*, and *Tfap2*) or neural crest specification genes [59,65,160]. While these eye malformations are within the neural epithelial-derived optic cup, it is clear from expression analysis and conditional knockout models that the neural crest cells within the periocular mesenchyme have cell non-autonomous effects on the optic cup. The neural crest cells restrict hedgehog signaling within the optic stalk and cup, which is critical for proper patterning of the retina [81,82,217]. Failure to establish the dorsal-ventral axis of the retina impairs normal optic cup growth and prevents ocular fissure closure. However, additional studies specifically assessing these interactions between periocular neural crest cells and the optic cup are required to fully understand the underlying mechanisms.

In addition, genetic disruption of later neural crest cell migration and differentiation can have significant effects on eye development. The *Pitx2* and *Foxc1* genes are the most well-studied as human autosomal dominant mutations in either is more often associated with Axenfeld–Rieger Syndrome and less commonly Peters Anomaly [9,184,206]. Although on rare occasions these diseases can be accompanied by optic nerve abnormalities such as Morning Glory Disc and colobomas, the majority of cases are limited to the anterior segment. However, gene knockout and knockdown animal models exhibit optic cup abnormalities. This discrepancy may be due to greater sensitivity of gene dosage effects on optic cup patterning compared to ocular neural crest cell migration and differentiation. Nevertheless, *Pitx2* and *Foxc1* are key regulators of eye development, and further studies are required to identify downstream targets both within the optic cup and neural crest cells.

While congenital eye anomalies are overall rarely occurring in approximately 1:5000 to 1:10,000 live births, understanding the genetics regulating neural crest cell development and ultimately the pathogenesis of these diseases is important [218]. Insight gained from improving our knowledge on neural crest genetics will lead to breakthroughs in stem cell and gene therapy treatments for these potentially blinding diseases.

## Figures and Tables

**Figure 1 jdb-08-00026-f001:**
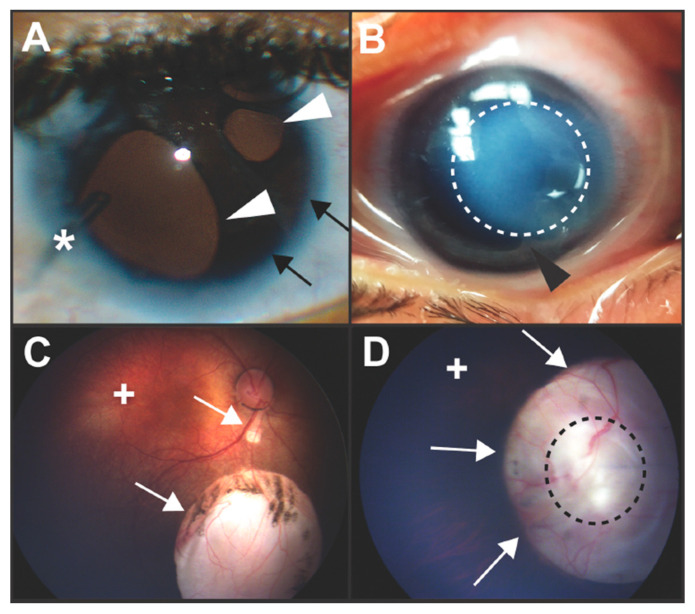
Clinical images of congenital ocular anomalies. (**A**) Axenfeld–Rieger syndrome is characterized by Rieger Anomaly (iris hypoplasia resulting in pseudopolycoria and corectopia (white arrowheads)) and Axenfeld Anomaly (anteriorization of Schwalbe’s line of the cornea (posterior embryotoxon, black arrows) with iris adhesions). These defects are due to abnormal migration and differentiation of neural crest cells into the anterior segment of the eye. Over 50% of individuals with Axenfeld–Rieger syndrome develop glaucoma which often requires surgery such as placement of a glaucoma drainage device (asterisk). (**B**) In Peters anomaly, there is a circumscribed central corneal opacification (outlined by dotted white line) with iris-corneal adhesions (black arrowhead). These anomalies are due to abnormal separation of the lens vesicle from the surface ectoderm resulting in absence of Descemet’s membrane and disruption of neural crest cell migration into the anterior segment. (**C**,**D**) Colobomas are due to incomplete closure of the ocular fissure and can affect the iris, zonules, retina, choroid, and optic nerve. Chorioretinal colobomas are inferior to the optic nerve and are characterized by an area that is devoid of retina and choroid (**C**, white arrows). In these types of coloboma, the macula (+) which accounts for central vision, is typically not affected. Optic nerve colobomas (**D**, white arrows) can cause severe vision loss especially if the entire optic nerve (outlined by black dotted line) is involved. Although the macula (+) may not be affected, the loss of the ganglion cell axons that comprise the optic nerve limits vision.

**Figure 2 jdb-08-00026-f002:**
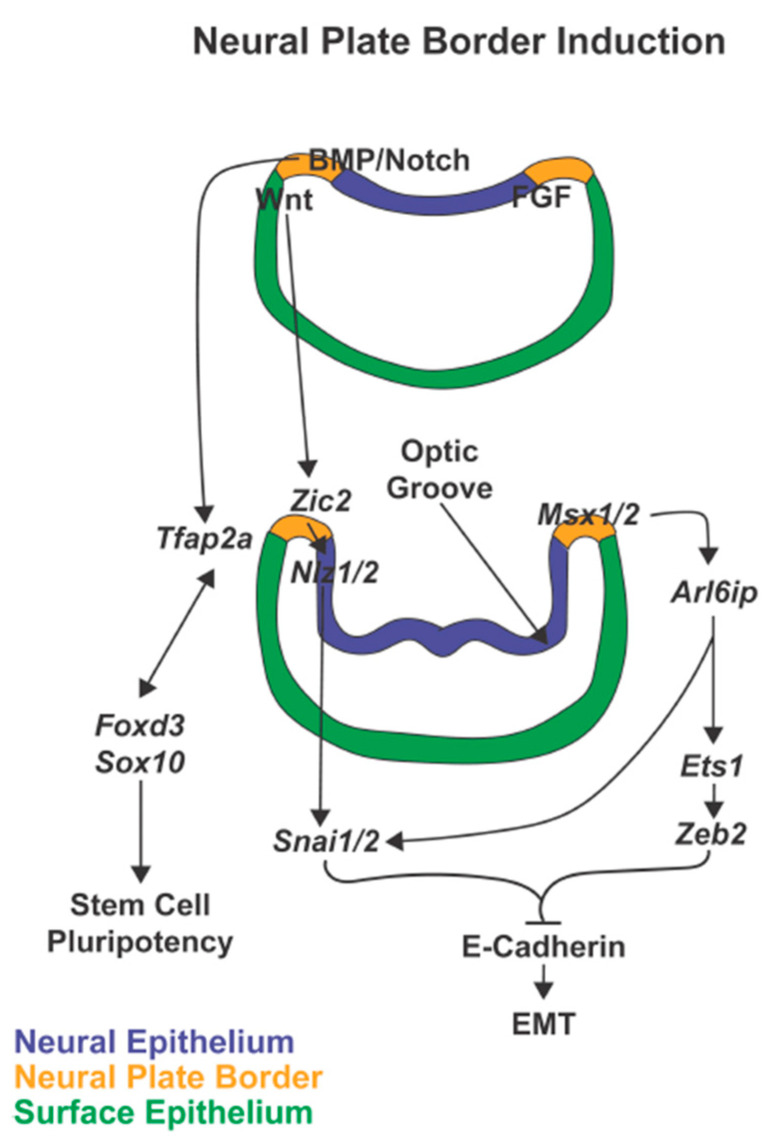
Neural plate border induction. In neural plate border induction, FGF followed by Notch and BMP signaling are expressed in the neural ectoderm while Wnt is expressed in the non-neural ectoderm. In the next phases of neurulation and optic groove formation, these signals then induce expression of the neural plate border specifiers, *Msx1/2, Zic2,* and *Tfap2.* These transcription factors, expressed within the neural plate border, trigger signaling cascades that maintain stem cell pluripotency and prepare the premigratory neural crest cells for epithelial-mesenchymal transition (EMT).

**Figure 3 jdb-08-00026-f003:**
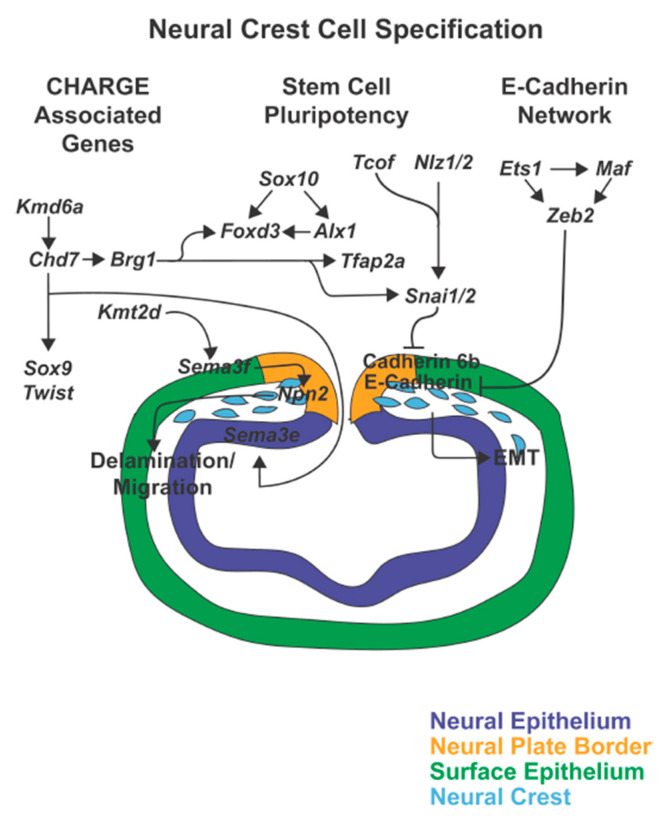
Neural crest cell specification. As the neural tube closes, genes including *Sox10*, *Foxd3*, *Alx1*, Tfap2a, *Tcof*, *Nlz1/2*, and *Snai1/2* induce neural crest cell identity and maintain stem cell pluripotency. Simultaneously, *Tcof*, *Nlz1/2*, and *Sna1/2*, together with *Ets1*, *Maf*, and *Zeb2* inhibit E-Cadherin and Cadherin 6b within the premigratory neural crest to trigger epithelial-mesenchymal transition (EMT). Furthermore, the CHARGE associated genes, *Chd7*, *Kmd6a*, Brg1, and *Kmt2d* regulate neural crest cell delamination and initiation of migration.

**Figure 4 jdb-08-00026-f004:**
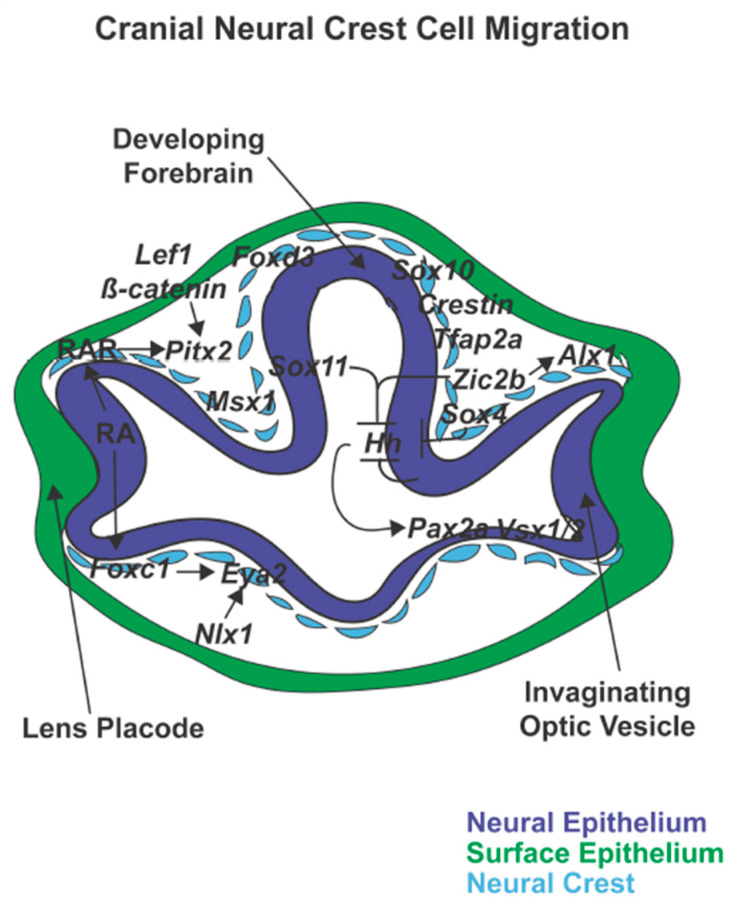
Cranial neural crest cell migration. Following closure of the neural tube, cranial neural crest cells expressing *foxd3*, *sox10*, and *crestin* migrate between the neural epithelial-derived forebrain and optic vesicle and the surface epithelium. *Sox4* within neural crest cells together with *Sox11* and *Zic2* regulate *hh-*patterning of the dorsal-ventral retina and subsequent expression of *Pax2a* and *Vsx1/2*. In addition, retinoic acid (RA) regulation of *Pitx2* and *Foxc1* within periocular neural crest cells is important for optic cup and anterior segment development.

**Figure 5 jdb-08-00026-f005:**
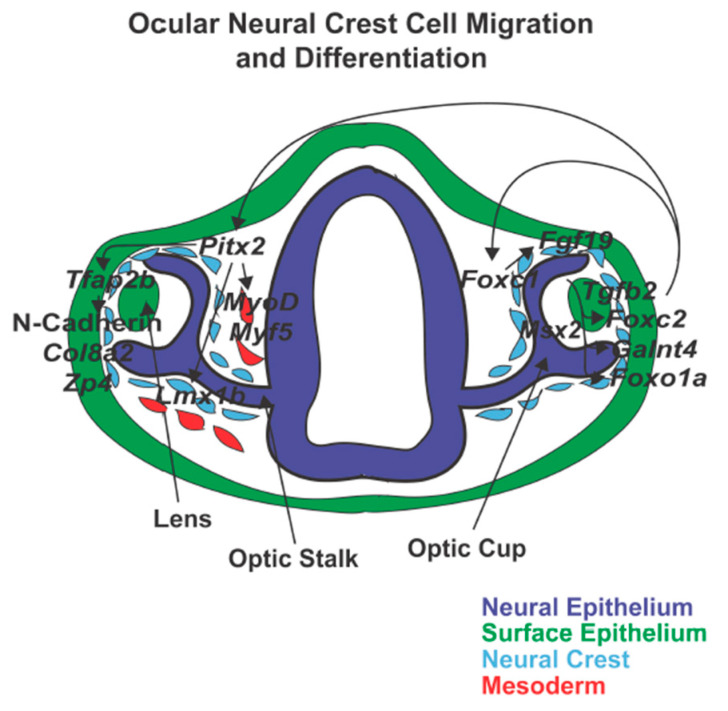
Ocular neural crest cell migration and differentiation. Neural crest cells surround the developing eye and enter into the ocular anterior segment between the surface epithelium and optic cup and through the ocular fissure. *Pitx2* is important for corneal and iridocorneal angle development and function through the regulation of *Tfap2b* and *Lmx1b.* In addition, *Pitx2* targets mesodermal cells and induces expression of *MyoD* and *Myf5* which triggers extraocular muscle formation. *Foxc1* also regulates anterior segment development by targeting *Fgf19, Foxc2, GaInt4,* and *Foxo1a*.

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
