# Peer review of "Genetics Underlying the Interactions between Neural Crest Cells and Eye Development"

_jdb, 2020, doi:10.3390/jdb8040026_

Round 1

Reviewer 1 Report

This manuscript is a very useful review of genetic networks governing neural crest development, and their especially involvement in eye development and disease.  The authors have carried out a good dissection of transcriptional regulatory networks, integrating clinical and model organism genetics and embryology.  Overall, the review article is timely and valuable and underscores importance of this understudied area.

Specific comments:

1. Comparative discussions of neural crest migration in mouse, chick, fish, and frog were very useful. For zebrafish, however, there should be a citation of a recent publication in PLoS Genetics (Takamiya et al., 2020; https://doi.org/10.1371/journal.pgen.1008774); this publication includes analyses of waves of neural crest migration in zebrafish.

2. Lines 115-117, summary of early neural plate border induction: while not the focus of this review, it would be helpful to include some of the primary literature as references, to complement the review articles that are currently referenced.

3. Line 245-247, “Further, loss of Kmt2d leads to decreased Sema3F, the secreted ligand for the Neuropilin2 (Npn2) receptor, which functions in migrating neural crest cells.” This sentence is a bit confusing: are the authors suggesting that neural crest cells have both Sema3F and Npn2?  That is, is Npn2 functioning in neural crest cells while these cells are also expressing Sema3F?

4. Line 356, “autosomal dominant mutations in human FOXD3”: it would be helpful to clarify what these alleles are. Are they dominant or haploinsufficient?  This would be useful to give perspective on how to compare this to the model organism knockout and knockdown experiments, which are primarily loss-of-function approaches.

5. Lines 435-437, “…knockout of Pitx2 also causes incompetence of the ocular neural crest cells to properly respond to signals from the surface-ectoderm-derived lens and corneal epithelium that activate corneal stromal and endothelial differentiation programs [181].” It is not clear that this particular reference (Asai-Coakwell et al., 2006) demonstrates this, as this study primarily involves measurements of central corneal thickness.

6. Lines 491-492, “Eya2 expression is promoted by foxc1a as well as nlx1 and serves to remodel the periocular mesenchyme in zebrafish.” The wording here is a bit confusing, as it sounds like the expression of Eya2 remodels the periocular mesenchyme.  Do the downstream targets remodel the cells, or the cellular environments?

7. The manuscript could use a thorough proofread. Here is a list of some of the minor errors:

-  Line 64-65, “… migrate to different regions of the embryo to give raise to a broad range of tissues…”  The word “raise” should be “rise”.

-  Figure 3, in the “Maintaining Pluripotency” box, “Fxod3” should probably be “Foxd3”.

-  Line 167, “However, their primary roll, as evidence by human pathology…”  The word “roll” should be “role”.

-  Line 180-182, “in zebrafish” is included twice in this sentence.

-  Line 260 and 262, the word “adheren” should be “adherens”.

-  Line 264, “…allowing it to accumulate in the nuclease and activate transcription…”  The word “nuclease” should be “nucleus”.

-  Line 318, “specifity” should be “specificity”.

-  Line 322, there are two commas after “Foxd3”.

-  Line 339, “morpholine” should be “morpholino”.

-  Lines 438-440, “RA, which is produced by the dorsal and ventral retina, but targets the periocular mesenchyme via RA receptors (RAR) alpha and gamma.”  The grammar should be corrected in this sentence (it does not appear to be a complete sentence).

-  Lines 473, “In Foxc1 knockout mice, the corneal is thickened and disorganized…”  The word “corneal” might be edited to “cornea”.

-  Line 475, “Knockdown of the zebrafish homologues Foxc1a…”  The word “homologues” should be “homologue”.

-  Line 497, “In mice periocular mesenchyme…”  The word “mice” could be edited to be “mouse”.

-  Lines 525-527, “However, additional studies specifically assessing these interactions…”  This sentence appears to be a fragment.

Author Response

1. Comparative discussions of neural crest migration in mouse, chick, fish, and frog were very useful. For zebrafish, however, there should be a citation of a recent publication in PLoS Genetics (Takamiya et al., 2020; https://doi.org/10.1371/journal.pgen.1008774); this publication includes analyses of waves of neural crest migration in zebrafish.

Response to the reviewer: Thank you for the suggestion of the paper. We added the reference to the paper and expanded the discussion on the cranial neural crest migratory waves in zebrafish.

2. Lines 115-117, summary of early neural plate border induction: while not the focus of this review, it would be helpful to include some of the primary literature as references, to complement the review articles that are currently referenced.

Response to the reviewer: We added additional references for primary literature on early neural plate border induction as suggested.

3. Line 245-247, “Further, loss of Kmt2d leads to decreased Sema3F, the secreted ligand for the Neuropilin2 (Npn2) receptor, which functions in migrating neural crest cells.” This sentence is a bit confusing: are the authors suggesting that neural crest cells have both Sema3F and Npn2?  That is, is Npn2 functioning in neural crest cells while these cells are also expressing Sema3F?

Response to the reviewer: The sentence was clarified and additional references were added. 

4. Line 356, “autosomal dominant mutations in human FOXD3”: it would be helpful to clarify what these alleles are. Are they dominant or haploinsufficient?  This would be useful to give perspective on how to compare this to the model organism knockout and knockdown experiments, which are primarily loss-of-function approaches.

Response to the reviewer: The Foxd3 variant is within the promotor and increases transcription in vitro. This is likely why the animal models do not phenocopy the human disease. This information was added along with corresponding references. 

5. Lines 435-437, “…knockout of Pitx2 also causes incompetence of the ocular neural crest cells to properly respond to signals from the surface-ectoderm-derived lens and corneal epithelium that activate corneal stromal and endothelial differentiation programs [181].” It is not clear that this particular reference (Asai-Coakwell et al., 2006) demonstrates this, as this study primarily involves measurements of central corneal thickness.

Response to the reviewer: The sentence was revised and additional references were added

6. Lines 491-492, “Eya2 expression is promoted by foxc1a as well as nlx1 and serves to remodel the periocular mesenchyme in zebrafish.” The wording here is a bit confusing, as it sounds like the expression of Eya2 remodels the periocular mesenchyme.  Do the downstream targets remodel the cells, or the cellular environments?

Response to the reviewer: This sentence was clarified to state that Eya2 induces apoptosis which results in remodeling of the periocular mesenchyme. 

7. The manuscript could use a thorough proofread. Here is a list of some of the minor errors:

-  Line 64-65, “… migrate to different regions of the embryo to give raise to a broad range of tissues…”  The word “raise” should be “rise”.

-  Figure 3, in the “Maintaining Pluripotency” box, “Fxod3” should probably be “Foxd3”.

-  Line 167, “However, their primary roll, as evidence by human pathology…”  The word “roll” should be “role”.

-  Line 180-182, “in zebrafish” is included twice in this sentence.

-  Line 260 and 262, the word “adheren” should be “adherens”.

-  Line 264, “…allowing it to accumulate in the nuclease and activate transcription…”  The word “nuclease” should be “nucleus”.

-  Line 318, “specifity” should be “specificity”.

-  Line 322, there are two commas after “Foxd3”.

-  Line 339, “morpholine” should be “morpholino”.

-  Lines 438-440, “RA, which is produced by the dorsal and ventral retina, but targets the periocular mesenchyme via RA receptors (RAR) alpha and gamma.”  The grammar should be corrected in this sentence (it does not appear to be a complete sentence).

-  Lines 473, “In Foxc1 knockout mice, the corneal is thickened and disorganized…”  The word “corneal” might be edited to “cornea”.

-  Line 475, “Knockdown of the zebrafish homologues Foxc1a…”  The word “homologues” should be “homologue”.

-  Line 497, “In mice periocular mesenchyme…”  The word “mice” could be edited to be “mouse”.

-  Lines 525-527, “However, additional studies specifically assessing these interactions…”  This sentence appears to be a fragment.

Response to the reviewer: Thank you for detecting these errors. They were corrected and we proofread the paper again. 

Reviewer 2 Report

The review: "Genetics underlying the interactions between the 2 neural crest and eye development" is an enumeration of genes expressed by neural crest cells at different developmental times and of their roles in eye development.

While the review is covering well the role of NCCs in the eye itself I suggest that the signalling role of NCCs for the formation of extraocular muscles and its contribution to periocular mesenchyme should also be included. After all the the development of the eye goes hand in hand with the development of the muscles which control its movements. In particular I would suggest to speak of FoxL2 in cranial NCCs (see for example Heude et al. 2015) and the disease Blepharophilosis, ptosis, epicantus inversus syndrome. The importance of NCCs in the periocular mesenchyme could also be seen, for example in the preprint https://doi.org/10.1101/2020.01.07.897694 which cites several other interesting papers.

Minor points:

In the title, introduction and several points of the paper "Neural Crest" should be changed by "Neural Crest Cells" and possibly abbreviated as NCCs. Indeed the "Neural Crest" is a structure of the embryo while the authors refer almost invariably to "Neural Crest Cells" which are cell migrating in the embryo and contributing to eye formation.

It is important to revise the quality of the English presentation.

I think that including one or more summary diagrams showing how the expression of different genes is associated to specific developmental process in the eye could help the reader to understand the role of individual genes.

Author Response

While the review is covering well the role of NCCs in the eye itself I suggest that the signalling role of NCCs for the formation of extraocular muscles and its contribution to periocular mesenchyme should also be included. After all the the development of the eye goes hand in hand with the development of the muscles which control its movements. In particular I would suggest to speak of FoxL2 in cranial NCCs (see for example Heude et al. 2015) and the disease Blepharophilosis, ptosis, epicantus inversus syndrome. The importance of NCCs in the periocular mesenchyme could also be seen, for example in the preprint https://doi.org/10.1101/2020.01.07.897694 which cites several other interesting papers.

Response to the reviewer: We agree that neural crest cells are required for extraocular muscle development. Information incorporating pitx2 and foxl2 and their roles in extra ocular muscle development was added. 

Minor points:

In the title, introduction and several points of the paper "Neural Crest" should be changed by "Neural Crest Cells" and possibly abbreviated as NCCs. Indeed the "Neural Crest" is a structure of the embryo while the authors refer almost invariably to "Neural Crest Cells" which are cell migrating in the embryo and contributing to eye formation.

Response to the reviewer: We made the changes as suggested throughout the document.

It is important to revise the quality of the English presentation.

Response to the reviewer: We proofread the paper and improved the English quality.

I think that including one or more summary diagrams showing how the expression of different genes is associated to specific developmental process in the eye could help the reader to understand the role of individual genes.

Response to the reviewer: Figures 2 and 3 were replaced with new Figures 2-5 which bring together the genes and signaling pathways at different stages of neural crest cell and eye development. 

Round 2

Reviewer 2 Report

The authors have considerably improved this paper which, in my view, can now be published.